# Target Span Detection for Implicit Harmful Content

## ABSTRACT

Identifying the targets of hate speech is a crucial step in grasping the nature of such speech and, ultimately, in improving the detection of offensive posts on online forums. Much harmful content on online platforms uses implicit language – especially when targeting vulnerable and protected groups – such as using stereotypical characteristics instead of explicit target names, making it harder to detect and mitigate the language. In this study, we focus on identifying implied targets of hate speech, essential for recognizing subtler hate speech and enhancing the detection of harmful content on digital platforms. We define a new task aimed at identifying the targets even when they are not explicitly stated. To address that task, we collect and annotate target spans in three prominent implicit hate speech datasets: SBIC, DynaHate, and IHC. We call the resulting merged collection Implicit-Target-Span. The collection is achieved using an innovative pooling method with matching scores based on human annotations and Large Language Models (LLMs). Our experiments indicate that Implicit-Target-Span provides a challenging test bed for target span detection methods.

## CCS CONCEPTS

• **Hate Speech** → **Target Classification**; • **Pooling**;

## KEYWORDS

Hate speech detection, Pooling, Dataset

**ACM Reference Format:**
Anonymous Author(s). 2018. Target Span Detection for Implicit Harmful Content. In *Woodstock '18: ACM Symposium on Neural Gaze Detection, June 03–05, 2018, Woodstock, NY*. ACM, New York, NY, USA, 5 pages. https://doi.org/XXXXXXX.XXXXXXX

## 1 INTRODUCTION

Social media provides a space where individuals can share their views openly and with a degree of anonymity. Yet, this has also led to the misuse of these platforms for disseminating harmful materials, including hate speech and toxic or offensive language. Numerous scholars are developing automated systems to identify and reduce the spread of such harmful materials. The majority of these methods rely on supervised classification methods [13, 21, 22], which are usually black boxes [1, 2, 7]. These black box systems cause their classifications into the binary *offensive* or *not* to be less explainable for humans as to why certain content should be removed from platforms [11]. When content is classified as *offensive*

**Content:**

> "Songwriters don't belong and never will, so let's just remove the piano brains from this place!"

***Implicit Target Span Identifier* Output:**

Target Spans: Songwriters, piano brains

**Figure 1: Example of target span detection from a hate speech dataset.**

or *not*, without any explanation, it leaves people unsure why the content should be removed [11]. One of the important aspects of explainable hate-speech detection is identifying the targeted groups in the given content, the focus of this work.

Although the literature has explored identifying hate speech targets using clear identifiers like actual names and visible traits, the implicit targeting of protected groups has received less attention. The task is challenging due to indirectness, metaphors, and stereotypes [9]. This phenomenon may only be perceptible and recognizable by the community that is targeted due to its subtlety and target-specific nuances, thereby remaining undetected by automated systems and other individuals [15]. It is crucial to detect the targeted spans in a context due to the need for interpretable and explainable hate speech models in this area. One of the applications of target span detection is to improve the explainability and interpretability of hate speech models by providing information about the words or phrases that helped with the model decision. Moreover, annotating and training target span detection datasets and models can contribute to the limited vocabulary in this area to identify different aspects of implicit hate speech in context [1]. To expand research in this area, we introduce the *implicit* Target Span Identification task (iTSI), which requires a model to identify target spans in harmful content, whether explicit or implicit. An example of this task is given in Figure 1.

To study the iTSI task we need an annotated training dataset. A few earlier studies introduced hate-speech datasets that also identify the target in the content. For instance, HateXplain [12] is a hate-speech dataset with the names of the targeted groups identified. Calabrese et al. [3] published a dataset with annotated spans for target, derogation, protected characteristics, etc. Barbarestani et al. [1] provided an annotation mechanism for highlighting target spans in toxic language. However, none of these studies annotated target spans that are *implicit*, focusing only on those that are explicitly presented by name.

In this study, we introduce a challenging dataset, Implicit-Target-Span (ITS), that captures both implicit *and* explicit target spans. It is built on three existing implicit hate speech datasets: Dyna-Hate [19], IHC [8], and SBIC [17]. To avoid the cost of exhaustive manual annotation, we explore a novel blending of human and LLM techniques. We ask annotators to tag a small portion of the dataset and then use LLMs through *pooling* technique [10] popularized by TREC [20] to choose the closest strategy to the human performance

using a new evaluation metric. Our new ITS dataset comprises 57k annotated samples with an average number of around 1.7 target spans (targets of hate speech) per sample. Although previous IHC and SBIC datasets, identify target groups, they do not highlight any text span. ITS enhances this by marking both explicit and implicit references to targets. For instance, in Figure 1 "songwriters" is an explicit target, while "piano brains" is an implicit reference to songwriters. We discovered approximately 19,000 unique targets across IHC and SBIC datasets, a significant increase from the earlier figure of about 1,000 targets in the original dataset implying that there are nearly 20 times more implicit targets than before.

We establish a performance baseline on our dataset using a BIO tagging approach [16]. We evaluate this baseline on various transformer-based encoders. Our results and analysis show the importance of iTSI and the new dataset to support significant research on target span identification in implicit hate speech. Our contributions include (1) introducing and formalizing a new target span identification task (iTSI), (2) a novel pooling-based annotation protocol to annotate both implicit and explicit target spans to create ITS, (3) establishing strong baselines on ITS, and (4) identifying challenges and suggesting improvements in building future approaches for this task.

## 2 IMPLICIT TARGET SPAN IDENTIFICATION

Given content $C$ represented as a sequence of tokens $C = [t_1, \ldots, t_n]$, where $t_i$ denotes the $i$-th token and $n$ is the total number of tokens in the content, the iTSI task is to identify token spans within $C$ that target a protected group. The output of this system is a set of non-overlapping tuples $S$, where each tuple $(s_{si}, s_{ei})$ corresponds to the start and end indices of a detected target span within $C$. Thus, $S = \{(s_{s1}, s_{e1}), (s_{s2}, s_{e2}), \ldots, (s_{sk}, s_{ek})\}$, with $s_{si}$ and $s_{ei}$ denotes the start and end token indices of the $i$-th detected target span, and $k$ represents the total number of detected target spans. The function $f(C) \rightarrow S$ maps the content $C$ to the set of target spans $S$, identifying the token sequences within $C$ that target a protected group.

For evaluating a model's ability to correctly identify target spans, the ground truth is similarly denoted as a set of non-overlapping tuples representing the actual start and end indices of target token spans within the content, $G = \{(g_{s1}, g_{e1}), (g_{s2}, g_{e2}), \ldots, (g_{sm}, g_{em})\}$ where $m$ is the number of ground truth target spans. By comparing the output $S$ with $G$, we evaluate the performance of the model $f$.

## 3 DATASET CONSTRUCTION

We have developed ITS, a dataset specifically for highlighting implicit target spans (as well as explicit targets), utilizing a method inspired by pooling. Our collected dataset consists of a total of 57k annotated samples. Statistics of ITS are given in Table 1.

To reduce the reliance on prompt and the LLM for weakly labeling the dataset, we utilize a pooling method to gather annotated target spans in an implicit hate speech corpus. This involves employing an array of high-quality Large Language Models (LLMs) with a handful of prompts that were found to be effective for this task and comparing their performance with human annotators to find the best prompt and LLM. We determine the most effective LLM and prompt combination by using our $F1_M$ metric, an F1 score

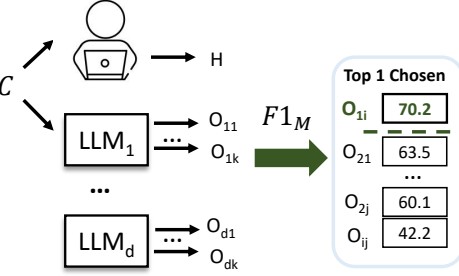

**Figure 2: The framework for choosing the best annotation strategy among different LLMs using a comparison with human annotator and $F1_M$ score. The strategy with the highest score is chosen.**

of the *precision* and *recall* of the generated targets in terms of their adherence (match) to target spans previously annotated by human reviewers.

After identifying the most effective combination of LLM and prompt, we proceed to use the selected pair to annotate the entire dataset. An overview of the process is given in Figure 2.

*Definition.* Let $C$ be the content that needs to be annotated, and $G$ be the ground truth target spans (section 2). Output target spans from the LLM are defined as $O = \{(o_{s1}, o_{e1}), \ldots, (o_{sn}, o_{en})\}$ with corresponding tokens for the $i$th span in $O$ defined as $T_{Oi} = \{t_{o_{si}}, \ldots, t_{o_{ei}}\}$ similarly for each $i$th span (of $m$) in $G$ we have $T_{Gi} = \{t_{g_{si}}, \ldots, t_{g_{ei}}\}$

$F1_M$ is defined as the harmonic mean of $Rec_M$ and $Prec_M$, where:

$$Rec_M = \frac{1}{|G|} \sum_{i=1}^{m} PM_r(i) \quad \text{and} \quad Prec_M = \frac{1}{|O|} \sum_{i=1}^{n} PM_p(i) \quad (1)$$

*Recall* based partial match score is the proportion of the ground truth covered by the output:

$$PM_r(i) = \begin{cases} 1 & \text{if } (g_{si}, g_{ei}) \in O \\ \frac{T_{Oj} \cap T_{Gi}}{|T_{Gi}|} & \text{if } \exists (o_{sj}, o_{ej}) \text{ s.t. } o_{sj} \geq g_{si} \ \& \ o_{ej} \leq g_{ei} \\ 0 & \text{otherwise} \end{cases} \quad (2)$$

and *precision* based partial match score ($PM_p$) is defined similarly as the proportion of the output that matches ground truth. $PM_p$ and $PM_r$ thus calculate the scores for both exact and *partial* matches. A partial match occurs for $PM_r$ if there is an output span that is contained partially within a ground truth span – and correspondingly for $PM_p$, a partial match occurs if a ground truth span is contained partially in the output span.

*Human Annotation.* Hate speech, even when implicit, can be troubling to encounter, so we drew three annotators, $A_1$, $A_2$, and $A_3$, from researchers involved in this project. They were tasked with marking up $N = 50$ samples that were randomly selected across the three datasets, distributed so the sub-collections were roughly balanced and so that targets were also roughly balanced.

The process began with a thorough briefing on the task, followed by each annotator receiving the same $N$ samples and guidelines. The annotators then highlighted explicit and implicit target spans in each of the samples. The collective responses from all annotators were then compiled into the union of the highlighted spans, where

| Fold | IHC | | | DynaHate | | | SBIC | | |
|------|---------|-----------|-----------|---------|-----------|-----------|---------|-----------|-----------|
| | Samples | TPC | ALT | Samples | TPC | ALT | Samples | TPC | ALT |
| Train | 5087 | 1.7 (0.9) | 1.5 (0.7) | 17799 | 1.6 (2.0) | 1.4 (0.7) | 22019 | 1.4 (0.8) | 1.3 (0.8) |
| Dev | 636 | 1.7 (0.9) | 1.5 (0.6) | 2190 | 1.5 (1.0) | 1.4 ( 0.6) | 3235 | 1.4 (0.8) | 1.3 (0.7) |
| Test | 636 | 1.7 (0.9) | 1.5 (0.7) | 2186 | 1.5 (0.9) | 1.4 (0.6) | 3339 | 1.4 (0.8) | 1.3 (0.7) |
| Total | 6359 | 1.7 (0.9) | 1.5 (0.7) | 22175 | 1.6 (1.1) | 1.4 (0.6) | 28593 | 1.4 (0.8) | 1.3 (0.8) |

**Table 1: Statistics of the annotated dataset. TPC is the average number of Target spans Per Content and ALT is the Average Length of Targets in content (number of tokens) The standard deviation is given in parentheses for each metric. We used the original train/dev/test distribution for the SBIC dataset and for IHC and DynaHate we randomly split the available dataset with an 8/1/1 ratio.**

| System | Prompt 1 | Prompt 2 | Human Inst |
|--------|----------|----------|------------|
| GPT-3.5 | 70.2 | 63.5 | - |
| Vicuna-13B | 63.0 | 61.0 | - |
| Llama2-13B | 61.8 | 60.1 | - |
| phi-2 | 42.2 | 38.0 | - |
| $A_1$ | - | - | 71.0 |
| $A_2$ | - | - | 62.3 |
| $A_3$ | - | - | 52.0 |

**Table 2: Comparison between different LLMs based on $F1_M$ (percent). The results are the average $F1_M$ scores across the samples.**

| Agreement | $A_1\&A_2$ | $A_1\&A_3$ | $A_2\&A_3$ |
|-----------|-----------|-----------|-----------|
| DSC | 61.0 | 56.0 | 52.0 |
| LCS | 84.8 | 86.2 | 85.0 |

**Table 3: Inter-annotator agreement on pairwise comparisons.**

overlapping responses were extended to the largest span that encompassed them all. The level of agreement among the annotators before combining their inputs is detailed in Table 3. We use the Longest Common Subsequence (LCS) and Dice Coefficient (DSC) as metrics for measuring the inter-annotator agreement. Similar to Calabrese et al. [3] we do not use Kohen's Kappa agreement in this study as we are evaluating the agreement on the sequence of tokens rather than individual tokens. Moreover, the task requires a partial match score as well as an exact match score which is not possible with Kappa agreement.

*Pooling Approach.* For selecting an optimal annotation system, we opt for $K$ prompts and $D$ LLMs as potential candidates. We aim to identify which LLM and prompt combination most closely aligns with human-level performance. To achieve this, we generate a pool of $K \times D$ outputs from these combinations and then rank the entire pool based on $F1_M$. The combination yielding the highest score is then chosen as the superior annotation system.

For this task we use four LLMs (GPT-3.5 turbo checkpoint, Vicuna-13B [5], Llama2-13B chat version [18] and phi-2[1]. To choose the best LLM and prompt[2], we compare the human-annotated target

[1]https://huggingface.co/microsoft/phi-2
[2]We release the code and data at https://suppressed.for.review.

spans for the same samples with the LLM outputs and rank the similarity of outputs to the aggregation of all the human responses using $F1_M$. The prompts used in the experiment are as follows:

**Prompt 1:** *Given the text highlight or underline parts of the text that mention or refer to the specific target. The target is sometimes not explicitly mentioned and you have to look for parts that implicitly refer to the target.*

**Prompt 2:** *The task is to highlight multiple text spans from the given input hate speech content that explicitly and/or implicitly mentions, refers to a specific protected group or their representation or characteristics that have been targeted.*

The output ranking scores are given in Table 2. As can be seen, GPT-3.5 achieves the highest score on Prompt 1 followed by Vicuna-13B which is an open source LLM. Overall we can see that all LLMs have closer performance to the aggregations of human annotators with prompt 1 than prompt 2. Additionally, we calculate the $F1_M$ of each human annotator ($A_i$) for aggregated annotations and observe that GPT-3.5 has a very close performance to $A_1$. We choose GPT-3.5 on prompt 1 for annotating the full dataset to create ITS.

## 4 TARGETDETECT: BASELINE MODEL

We adopt an existing model architecture to train ITS and evaluate it in in-domain and out-of-domain settings. We refer to this model as TargetDetect that takes content $C$ and predicts the target spans $S$. The TargetDetect is a sequence tagging framework, based on a pre-trained transformer-based encoder(e.g., BERT, RoBERTa, etc.). Each token within $C$ undergoes a classification process employing the BIO (Beginning, Inside, Outside) tagging scheme. This approach marks each token as the *Beginning* of a target span, *Inside* a span, or *Outside* the spans of interest. By adopting this tagging strategy, we train a model and experiment with different encoders to identify all the target spans in content. The training process for TargetDetect involves minimizing the error between the model's predictions $S$ with the correct answers $G$ using cross-entropy loss defined as $L_{CE} = -\sum_{i=1}^{N} y_i \log p_i$. $y_i$ denotes the ground truth label for the $i$-th token in the BIO tagging scheme, and $p_i$ is the predicted probability that the $i$-th token is the beginning of, inside, or outside a target span.

## 5 EXPERIMENTS

In our experiments, we evaluate the performance of TargetDetect on the cross-domain test dataset from our data collection (IHC, SBIC, and DynaHate) as well as PLEAD [3], a publicly available

| Encoders | $D_{train}$ | IHC | | | | DynaHate | | | | SBIC | | | |
|---|---|---|---|---|---|---|---|---|---|---|---|---|---|
| | | F1 | P | R | Acc | F1 | P | R | Acc | F1 | P | R | Acc |
| Bert-Base | in-Domain | 67.0 | 70.0 | 64.0 | 93.5 | 76.0 | 74.5 | 77.7 | 95.0 | 63.4 | 58.2 | 69.6 | 94.1 |
| | All | 69.0 | 67.4 | 70.5 | 92.6 | 76.8 | 76.9 | 76.7 | 95.0 | 71.1 | 72.1 | 70.1 | 94.5 |
| Hate-Bert | in-Domain | 65.0 | 68.5 | 61.6 | 92.5 | 76.4 | 76.2 | 76.7 | 95.0 | 71.0 | 72.0 | 70.0 | 94.1 |
| | All | 69.3 | 66.7 | 72.1 | 92.6 | 77.4 | 76.5 | 78.3 | 95.0 | 71.8 | 71.9 | 71.6 | 94.3 |
| RoBERTa-Large | in-Domain | 71.7 | 72.4 | 71.1 | 92.4 | 79.1 | 79.8 | 78.3 | 95.0 | 76.3 | 74.6 | **78.0** | 94.3 |
| | All | **76.5** | **74.4** | **78.7** | **93.0** | **80.8** | **79.8** | **81.7** | **95.1** | **77.4** | **77.1** | 77.7 | **94.5** |

**Table 4: Overall performance comparison between different encoders.** $D_{train}$ **refers to** *in-Domain* **training (on each train set separately) or training on** *All* **three train sets combined before testing on each set separately. The results are reported as the percentage of the indicated measure.**

| Training set | PLEAD | IHC | DynaHate | SBIC |
|---|---|---|---|---|
| PLEAD | 52.0 | 36.6 | 53.8 | 36.9 |
| IHC | 48.2 | 71.7 | 72.3 | 63.3 |
| DynaHate | 51.0 | 72.8 | 79.1 | 71.8 |
| SBIC | 48.0 | 74.4 | 78.0 | 76.3 |

**Table 5: The diagonal shows the results for full fine-tuning (in-domain training). We report results based on the F1 score.**

human-annotated dataset capturing a combination of explicit and implicit target spans. We use only "hate" annotated samples from PLEAD (similarly we filter our dataset). Even though contents may have multiple target spans, the original PLEAD dataset, published the original data as exactly one target span per content at a time. We reunite such samples to have multiple target spans, to have the same data distribution as ours. The PLEAD dataset is composed of 2,462 samples, including 1,969 for training, 246 for development, and 247 for testing. Average Target span Per Content (TPC) and Average length of targets (ALT) with standard deviations (cf. Table 1) are 1.9(0.9) and 1.6(0.7), respectively.

*Encoders.* To train TargetDetect we use three different transformer encoders: 1) Bert-Base [6]; 2) RoBERTa-Large [23]; and 3) Hate-BERT [4], a BERT model pre-trained on hate data.

*Evaluation Metrics.* Since we have trained a sequence tagging model target span detection, we use the seqeval [14] metric to evaluate different encoders. This metric calculates F1, Precision, Recall, and Accuracy based on the predicted tagging labels and ground truth labels for each token in the content.

## 5.1 Results

The overall token tagging classification results based on different encoders and test datasets are given in Table 4. The best performance is consistently achieved using RoBERTa-Large encoder. Among the encoders, Bert-Base has the worst performance and Hate-Bert performs slightly better with the average F1 score gain of 4% for test set performance in in-domain trained models and 1.5% for training on all the training sets at once. Overall we see that training in all

the domains consistently improves the performance compared to training on only an in-domain dataset.

We evaluate the effectiveness of training the baseline model on different training samples including our proposed dataset ITS collection and human annotated PLEAD dataset. Performance comparison of TargetDetect with RoBERTa-Large encoder on ITS and PLEAD is shown in Table 5. The diagonal shows the F1 score results for in-domain full finetuning whereas the rest are different combinations of training on one domain and testing in another (a zero-shot approach). RoBERTa-Large is used as an encoder in this experiment. The zero-shot experiment across ITS test sets results in a higher F1 score except when TargetDetect is trained on PLEAD. Training on ITS and testing on PLEAD results in only 5% performance loss on average in F1 score. The DynaHate-trained model has the least performance loss of only 2% (i.e., 51.0% with DynaHate-trained samples vs 52.0% with PLEAD-trained samples. This indicates that training target detection with ITS and testing it on external, human-annotated datasets can achieve performance similar to that of training on in-domain, human-annotated datasets, demonstrating ITS's effectiveness across various domains.

## 5.2 Error Analysis

We use RoBERTa-Large and IHC (finetuned on the IHC train set) for error analysis. From our quantitative analysis, we discovered that in 26.5% of the instances where errors occurred, at least one of the predicted spans within a piece of content exhibited a "boundary" error, meaning the predicted span partially overlaps with the actual span but falls short by several characters. For instance, whereas the correct span might be "European people," the model might only identify "European" however, in approximately 90% of such cases, the most important words or phrases to detect the targeted group were present. The boundary errors occurred in the least important words such as stop words.

Additionally, we observed that one of the common model failures is due to the discrepancy between the *number* of predicted spans and the number of ground truth spans. Considering the failed cases, roughly a third predicted too many spans, a third predicted too few, and a third predicted the right number. Missing or over-generating are both important sources of error, suggesting the need for improvement.

Our manual observation of the failed cases also showed interesting patterns that can be useful for future research. We identify a few main failure patterns: 1) **Obfuscated targets** where the model fails to correctly identify the target because the original word or phrase is obfuscated – e.g., "songwriter" is written as "s0ngwr1t3r"; and 2) **Implicit and subtle references to targets** where the trained model fails to recognize the target because lexically and semantically the original target is replaced with its indirect reference – e.g., hiding the "songwriter" by substituting it with "conservatory graduates".

## 6 CONCLUSION

In this paper, we introduce a reliable weak labeling approach using a novel pooling-based framework for the challenging implicit target span detection task iTSI. The generated dataset using this pooling approach will be released to enhance research in this area. We also introduce a baseline model, TargetDetect, to evaluate the task on ITS. Our results show that ITS trained model can achieve comparable performance to the model trained on the human-annotated dataset. We also show that ITS is a challenging dataset that presents several potential research directions.

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

Received 20 February 2007; revised 12 March 2009; accepted 5 June 2009