# OpenReview forum: "Target Span Detection for Implicit Harmful Content"
_ACM.org/SIGIR/ICTIR/2024/Conference — ICTIR 2024_

### Official Review · Reviewer_eZ1Z · 2024-05-13

**Rating:** 0
**Confidence:** 4

**Objective Part Of Review:**

In this paper the authors focus on identifying implied targets of hate speech, they define a new task, build a dataset and introduce a baseline method. The authors use human annotations for validation since they use a pooling approach for annotation.

The problem and the results are clearly stated.

One of the limitations of the paper is that there is not proper definition of the implicit and explicit target, the authors give an example but this is not enough. In general, in the introduction it was not clear what is the implicit target and why it is necessary to identify them given that the explicit targets are already annotated in other datasets.

Also, the human annotation is done by researchers involved in the study. However, it was mentioned that the task in complicated and that this “phenomenon may only be perceptible and recognizable by the community that is targeted”. Given those, I am wondering whether the researchers have the right expertise to do this annotation. It would make more sense to ask experts to do those annotations; also there are no details regarding the guidelines and how many annotations each of them did. Finally, given the complexity of the task, I would like to know the demographics of the experts.

Another question is regarding the prompts; did the authors run the prompts several times to see if the results change?

In table 1, it is useful to show the overall numbers as well.

**Subjective Part Of Review:**

The paper was easy to follow and understand. The main contribution of the paper is the development of a dataset on implicit target detection in hate speech. The authors also apply different baselines on the problem and evaluate their performance. Although the task is relevant and important, there are some limitations (see my previous comments) in the annotation process that need to be clarified.

---

### Official Review · Reviewer_s8SE · 2024-05-15

**Rating:** -2
**Confidence:** 5

**Objective Part Of Review:**

The paper introduces a new dataset of spans of explicit and implicit targets of harmful social media content. "Implicit targets" should be better defined. Section 2 has "Implicit" in the title, but it covers standard span annotation and evaluation of span detections.

The paper claims to contribute a new annotation approach based on Pooling using Large Language Models to carry out the annotation. The relation to Pooling is however not completely clear. The similarity to Pooling is clear as far as the use of four different LLMs with two different prompts to generate spans. The "outputs" of these models are then used, but Pooling uses ranked documents as input - it is not made clear at all how these spans are mapped to documents for Pooling. Furthermore, in Pooling the relevance judgements are done on the pool but in this approach, manual span annotation is necessary in order to create the pools. The links to Pooling are therefore rather tenuous.

The actual contribution of the paper is therefore the dataset. However, given the problems with the process used to create the dataset outlined above and that the ICTIR is not the correct conference to publish a dataset contribution, this paper is not suitable for the conference.

**Subjective Part Of Review:**

Based on the objective part of the review, I don't think that this paper should be accepted.

---

### Official Review · Reviewer_zoVP · 2024-05-16

**Rating:** 1
**Confidence:** 3

**Objective Part Of Review:**

The contribution is easy to read and well written. The idea presented is straightforward and useful for future work on the topic. The dataset that will be released can be of definite use to the scientific community. The preliminary experimental results, albeit based on very simple architectures, are encouraging. The way the dataset is annotated is questionable. The authors based their preliminary analysis on the use of GPT-3.5 alone. This is a strong limitation of the research. It would have been useful and more robust to use the similarities obtained with the human annotators to weight the model differently in a majority-voting phase between the LLM annotators. In any case, the work seems valid to me.

**Subjective Part Of Review:**

The contribution is a preliminary study that may open up the possibility for future work on the subject. Considering the release of the resource I find it acceptable at the conference although the annotation methodology could have been more robust.

---

### Official Review · Reviewer_FWE1 · 2024-05-20

**Rating:** -1
**Confidence:** 4

**Objective Part Of Review:**

The paper creates an dataset for detecting hate speech targets, which is often implicit in nature. The authors start with a human annotated dataset and then apply 0-shot LLMs to predict the taget entities.

The use of a manually annotated dataset to optimize the choice of an LLM and a prompt combination is expected to lead to heavy biases towards the manual annotation itself, and is less likely to unexplored territories. If we prefer the LLM the output of which matches maximally with what humans have annotated then how can it extend the ground-truth by suggesting new annotations?

**Subjective Part Of Review:**

The paper is generally well-written. However, there are several points that lack clarity. For example, it is not clear to me what proportion of the ground-truth actually expanded due to the use of LLMs to infer weak labels. Are these weak labels merged with the strong (human-created) ones?
A high value of an evaluation metric on an "expanded dataset" doesn't mean that a downstream model is good. All this means is that the in-domain and the out-domain sets aren't comparable because they use different evaluation sets.

It's a bit surprising that prompt-1 which has got no explicit mention of "hate" works better than prompt-2.
What's the basis of selecting two prompts that are quite different in its meaning? The second one explicitly states to find entities against which/whom hate is targeted.
It wasn't clear to me how is the notion of "protected group" defined or is even relevant to this work? In the example, is "Piano brain" a protected group?

---

### Meta-Review · Area_Chair_UCNX · 2024-05-30

**Recommendation:** Reject
**Confidence:** 5

**Metareview:**

The paper describes the development of a dataset on implicit target detection in hate speech and contains some preliminary results of some baselines.

I suggest to reject the paper for ICTIR 2024 for the following reasons:
- the reviewers raise important open questions with respect to the creation of the dataset
- ICTIR does not really have a dedicated resource track and the connection of the paper's topic to IR is not that strong
-> a more fitting venue would be some NLP-related conference with a resource focus or a resource track